# Interactions between *Malassezia* and New Therapeutic Agents in Atopic Dermatitis Affecting Skin Barrier and Inflammation in Recombinant Human Epidermis Model

**DOI:** 10.3390/ijms24076171

**Published:** 2023-03-24

**Authors:** Yu-Jin Lee, Caren Yassa, Song-Hee Park, Seo Won Song, Won Hee Jung, Yang Won Lee, Hoon Kang, Jung-Eun Kim

**Affiliations:** 1Department of Dermatology, Eunpyeong St. Mary’s Hospital, College of Medicine, The Catholic University of Korea, Seoul 03312, Republic of Korea; 2Department of Developmental and Cell Biology, University of California, Irvine, CA 92697-227589, USA; 3Department of Systems Biotechnology and Institute of Microbiomics, Chung-Ang University, Anseong 17546, Republic of Korea; 4Department of Dermatology, Konkuk University School of Medicine, Seoul 03312, Republic of Korea

**Keywords:** atopic dermatitis, *Malassezia*, anti-IL4Rα, ruxolitinib, reconstructed human epidermis model

## Abstract

Several studies have reported the pathogenic role of *Malassezia* in atopic dermatitis (AD); the significance of *Malassezia*’s influence on AD needs to be further investigated. Dupilumab, a monoclonal antibody to anti-Interleukin (IL) 4Rα, and ruxolitinib, a Janus kinase (JAK)1/2 inhibitor, are the first approved biologics and inhibitors widely used for AD treatment. In this study, we aimed to investigate how *Malassezia Restricta* (*M. restricta*) affects the skin barrier and inflammation in AD and interacts with the AD therapeutic agents ruxolitinib and anti-IL4Rα. To induce an in vitro AD model, a reconstructed human epidermis (RHE) was treated with IL-4 and IL-13. *M. restricta* was inoculated on the surface of RHE, and anti-IL4Rα or ruxolitinib was supplemented to model treated AD lesions. Histological and molecular analyses were performed. Skin barrier and ceramide-related molecules were downregulated by *M. restricta* and reverted by anti-IL4Rα and ruxolitinib. Antimicrobial peptides, VEGF, Th2-related, and JAK/STAT pathway molecules were upregulated by *M. restricta* and suppressed by anti-IL4Rα and ruxolitinib. These findings show that *M. restricta* aggravated skin barrier function and Th2 inflammation and decreased the efficacy of anti-IL4Rα and ruxolitinib.

## 1. Introduction

Atopic dermatitis (AD) is a chronic recurrent pruritic skin disease characterized by skin barrier dysfunction and an overactive T helper 2 type (Th2) immune response. 

One of the leading causes of damaged skin barrier in AD is the reduction of skin barrier-related molecules such as filaggrin (FLG), loricrin (LOR), and Involucrin (IVL) [1]. They form a dense network of keratin fibers to create a sturdy skin barrier, which provides mechanical strength and assists in controlling hydration [2]. In addition, a decrease in skin lipids also causes damage to the skin barrier. Ceramides are important lipids in the skin; they play a key role in maintaining the skin barrier [3]. Therefore, synthesizing ceramides and long-chain fatty acids is essential to maintain the lipid barrier. The level of CerS3 (ceramide synthase3), ELOVL1 (elongation of very long-chain fatty acid protein1), and ceramide are known to be decreased in the skin of AD patients [4,5]. Furthermore, the deterioration of the epidermal permeability barrier also causes skin barrier dysfunction. The corneodesmosome and intercellular lipid lamellae are the most crucial elements of the permeability barrier [6,7]. Lipid lamellae block the penetration of harmful substances into the intercellular space and regulate water loss. The corneodesmosome is the main intercellular adhesive structure of the stratum corneum (SC), making it a firm structure. The Th2 inflammatory response reduces the molecules and the integrity of the structures mentioned above in AD, which in turn damages the skin barrier.

Antimicrobial peptides (AMPs) are a class of peptides produced by skin cells and have antimicrobial properties [8]. AMPs play an important role in the innate immune response of the skin and have been implicated in the pathogenesis of AD. Th2 cytokines such as IL-4 and IL-13 have been found to downregulate the expression and activity of AMPs in the skin [9]. The downregulation of AMPs may impair the innate immune response, making skin more vulnerable to microbial and fungal infections in the AD patient. In recent years, studies on the correlation between AD and *Malassezia*, one of the skin commensal fungi, have been actively conducted [10].

*Malassezia* species are lipophilic yeasts, mainly colonized in the head and neck areas where skin lipid is rich [11]. *Malassezia* is recognized as a pathogen in various skin disorders, including seborrheic dermatitis, pityriasis versicolor, *Malassezia* folliculitis, and head and neck dermatitis in AD, and psoriasis, in immunologically competent hosts [12,13]. The cell wall components of *Malassezia* species bind to multiple receptors on the host cell membrane, resulting in ligand internalization and activation of the NLRP3 inflammasome as well as inflammatory signaling pathways, including the mitogen-activated protein kinase (MAPK), nuclear factor kappa B (NF-κB), and nuclear factor of activated T cells (NFAT) pathways [14,15]. For example, pathogen-associated molecular patterns (PAMPs) on the cell surface of *Malassezia* are recognized by toll-like receptors (TLRs) on host keratinocytes and activate inflammation. *Malassezia* species secrete proteases and toxic metabolites and generate reactive oxygen species (ROS) which contribute to the development of inflammation in keratinocytes through the NF-κB pathway [16]. The activated pathway can damage the skin barrier and cause secondary inflammation. *Malassezia* antigens, such as Mala s 1, can trigger an allergy response [17]. *Malassezia* species do not typically cause sensitization in healthy people, but a significant portion of AD patients show increased serum immunoglobulin E to *Malassezia* [15]. Among the 14 identified species of *Malassezia*, *M. sympodialis*, *M. globosa, M. furfur,* and *M. restricta* have been studied with respect to various human skin diseases [10,11,12,13,14,15,16,17,18]. *M. restricta, M. globosa,* and *M. sympodialis* are the most frequently identified species in AD skin [19]. However, there remain few research findings regarding their contribution to the pathogenesis of AD.

In AD, Th2-driving chemokines, such as TSLP, CCL17, and CCL22, are activated in response to allergens or other triggers, causing Th2 to release cytokines such as IL-4 and IL-13 [20]. These cytokines can induce the expression of proinflammatory cytokines and chemokines in keratinocytes, such as IL-1β, IL-6, and IL-8 [21]. They can lead to the recruitment of immune cells to the site of inflammation and exacerbation of AD symptoms. Th2 cytokines also contribute to skin barrier dysfunction by activating the Janus kinase-signal transducer and activator of transcription (JAK/STAT) pathway and reducing the production of skin barrier proteins [22,23,24]. 

Th1 cells produce interferon-gamma (IFN-γ) and other cytokines involved in cell-mediated immunity [25]. In AD, Th1 cells are known to be involved in chronic inflammation and tissue damage. High levels of IFN-γ have been found in the skin of chronic AD patients, and blocking this cytokine can improve AD symptoms [26]. Th17 cells produce IL-17 and other cytokines involved in the defense against extracellular bacteria and fungi [27,28]. Th17 cells are also present in the skin of AD patients, and their cytokines can stimulate keratinocytes to produce pro-inflammatory cytokines and chemokines, leading to skin inflammation and barrier dysfunction [29]. Overall, Th2 immune cells are still considered the primary drivers of AD, but Th1 and Th17 cells also play important roles in the chronic inflammation and tissue damage in AD [26,29].

Since Th2 immune cells play an essential role in developing and maintaining AD, drugs targeting Th2 immune response, such as IL-4, IL-13, TSLP, and JAK/STAT small molecules, have become an important strategy for developing new treatments for AD. Recently, IL-4 receptor alpha (IL-4Rα) and JAK/STAT-targeting AD therapies have been approved and used safely and effectively. IL-4Rα is a part of type II (IL-4Rα and IL-13Rα1) IL-4R receptor complexes, which activate downstream signaling pathways leading to the production of Th2 cytokines and recruitment of inflammatory cells to the skin [30]. Dupilumab (Dupixent^®^), a monoclonal antibody to anti-IL4Rα, is approved for treating moderate-to-severe AD, asthma, chronic rhinosinusitis with nasal polyposis, eosinophilic esophagitis, and prurigo nodularis [31]. Ruxolitinib (Jakafi^®^) is a selective JAK1 and JAK2 inhibitor used to treat myelofibrosis, polycythemia vera, and graft-versus-host diseases. Recently, the FDA approved ruxolitinib cream for the short-term treatment of mild-to-moderate AD [32]. Both ruxolitinib and anti-IL4Rα effectively block IL-4 and IL-13, key inflammatory cytokines in AD. Ruxolitinib and dupilumab are the first topical JAK inhibitors and biologics, respectively, approved by the FDA for treating AD, and are currently widely used in the clinic.

With the increasing use of the drug for AD treatment, a particular patient population was reported to experience an exacerbation of head and neck dermatitis or facial redness after using dupilumab [31,32]. Several case reports demonstrate that hypersensitivity to *Malassezia* species is the leading cause of these phenomena, and that antifungal agents were adequate for treatment [33]. However, whether *Malassezia* is the cause or result of AD flare-ups, whether these adverse effects result from interactions between the drugs and *Malassezia*, or how the drugs affect the skin in the presence of *Malassezia* are all understudied topics. Meanwhile, facial redness is not reported in AD patients treated with ruxolitinib. The pathogenesis and treatment methods of dupilumab-related facial redness vary depending on each individual; topical JAK inhibitors and stem cell therapy have been reported to be effective [34]. Nevertheless, the therapeutic mechanism of JAK inhibitors is still unknown. Therefore, as an extension of the clinical cases, we tried to study the pathomechanism interactions between *Malassezia* and anti-IL4Rα or JAK inhibitor. 

A growing interest in animal testing regulations and alternative models that reduce pain from a humanitarian perspective has replaced animal models with 3-dimensional (3D) human cell culture models [35]. RHE is one of the in vitro 3D models widely used in experimental settings. The use of RHE is essential for effective and novel AD treatment research. Since RHE has all epidermal skin layers, inoculation of *Malassezia* colonies on the surface of RHE can produce a biomimetic skin environment replicating physiological functions compared to a 2D keratinocyte culture model. As such, we used an RHE model to conduct experiments in an environment similar to in vivo human skin. We designed normal skin, AD-like skin (IL-4/IL-13-treated RHE),and treated AD lesion-like skin (anti- IL4Rα or ruxolitinib-treated IL-4/IL-13 RHE) using a 3D skin model. We divided each group into with or without MR inoculation. Then, we examined the role of *Malassezia* in each condition as well as the interactions between new therapeutic drugs and MR in each skin model. The control group refers to healthy skin without inflammation, and the IL-4/IL-13 group refers to AD-lesional skin. The groups treated with ruxolitinib or anti-IL4Rα in IL-4/IL-13 represent the AD skin treated with topical medication. Similarly, the MR-only group refers to healthy skin treated with MR only, and the MR-treated IL-4/IL-13 group refers to AD-induced skin with the presence of MR. The groups treated with ruxolitinib or anti-IL4Rα in MR-treated IL-4/IL-13 represent the AD skin topically treated with ruxolitinib or anti-IL4Rα in AD-induced skin with the presence of MR.

In this study, we investigate how *M. restricta* affects healthy skin, AD lesions, and AD lesions undergoing treatment with ruxolitinib or anti-IL4Rα in 3D RHE with respect to skin barrier function and inflammation. 

## 2. Results

### 2.1. Variation of Tissue Morphology of the AD-RHE Stimulated with Malassezia Restricta (MR)

First, we stained the RHE samples with H&E to observe the morphology after being stimulated with IL-4/IL-13 and/or MR (Figure 1). IL-4/IL-13 treatment resulted in severe hyperkeratosis, spongiosis, and acanthosis in the RHE samples. In addition, the treatment caused basket weave pattern hyperkeratosis in the SC layer, which became more dispersed. However, after treating with anti-IL4Rα and ruxolitinib, hyperkeratosis and spongiosis were significantly restored. MR treatment also caused loosely organized hyperkeratosis, acanthosis, and spongiosis in RHE compared to the control. Detachment of SC became more severe with MR-treated IL-4/IL-13 RHE compared to MR-untreated IL-4/IL-13 RHE. Even though spongiosis improved after receiving anti-IL4Rα and ruxolitinib treatment in MR-treated RHE, the deformation of the SC layer did not improve considerably. Overall, the skin layers of the MR-treated RHE were thicker than that of the MR-untreated RHE, and the SC layer structure was more dispersed and exfoliative in the MR-treated RHE than in the MR-untreated RHE.

### 2.2. Expression of Epidermal Skin Barrier and Antimicrobial Peptide-Related Molecules in AD-RHE Stimulated with Malassezia Restricta (MR)

We investigated skin barrier and AMP-related molecules in AD-RHE. The expression of skin barrier molecules was histologically investigated using immunofluorescence. FLG is expressed in the SC and stratum granulosum (SG) in RHE. In the MR-untreated RHE, the expression of FLG decreased when treated with IL-4/IL-13 and was moderately restored when treated with anti-IL4Rα or ruxolitinib (Figure 2A). IVL is expressed in SG and the upper stratum spinosum. In the MR-untreated RHE, the expression of IVL decreased when treated with IL-4/IL-13 and was also moderately restored when treated with anti-IL4Rα or ruxolitinib (Figure 2B). LOR is expressed in the SG of the epidermis. Similarly, in the MR-untreated RHE, the expression of LOR decreased when treated with IL-4/IL-13 and the same restorative pattern was shown when treated with anti-IL4Rα or ruxolitinib (Figure 2C). In the MR-only treated groups, only the expression levels of LOR decreased; the expression levels of FIG and IVL remained unchanged (Figure 2A–C). In the MR-treated IL-4/IL-13 RHE, the expression of barrier molecules was significantly reduced and was only partially restored by anti-IL4Rα or ruxolitinib treatment compared to MR-untreated IL-4/IL-13 RHE (Appendix A).

To further support the results above, quantitative protein analysis was conducted using Western blotting (Figure 2D,E). The protein levels of skin barrier molecules (FLG, IVL, LOR) were significantly decreased by IL-4/IL-13 treatment; the protein levels were restored by anti-IL4Rα and ruxolitinib treatments. In the MR-treated groups, MR exposure downregulated expression levels of LOR compared to the control; the expression levels were restored by anti-IL4Rα or ruxolitinib (Appendix A). FLG and IVL expressions are not affected by MR itself; however, MR inhibits the restorative effects of anti-IL4Rα and ruxolitinib on IVL expression in AD-RHE. Overall, the barrier-restorative effects of anti-IL4Rα and ruxolitinib on LOR and IVL were lower in the MR-treated group compared to the MR-untreated group.

The expression of AMPs was quantitatively investigated using Western blotting (Figure 2D,E). The protein level of LL-37 was not significantly changed when treated with IL-4/IL-13. However, when treated with anti-IL4Rα and ruxolitinib, the expression level increased significantly compared to the control. The protein level of β-defensin2 was increased when treated with IL-4/IL-13 and decreased when treated with anti-IL4Rα compared to the control. Ruxolitinib treatment did not reduce the level of β-defensin2. The production of LL-37 and β-defensin2 was significantly upregulated by MR (Figure 2E and Appendix A). The MR-treated IL-4/IL-13 group showed decreased levels of LL-37; the levels were upregulated by anti-IL4Rα and ruxolitinib treatments. There was not a significant difference shown between MR-treated groups in β-defensin2.

A damaged skin barrier is also associated with reduced ceramides with long-chain fatty acids in the SC. We examined the expression of CerS3 and ELOVL1 at the mRNA levels. As shown in Figure 2F,G, CerS3 and ELOVL1 were significantly downregulated by IL-4/IL-13 treatment and remarkably upregulated by additional anti-IL4Rα and ruxolitinib treatment. Compared to the control, CerS3 was downregulated by MR treatment, while ELOVL1 expression levels were not changed. However, CerS3 and ELOVL1 expressions were downregulated equally in the MR-treated IL-4/IL-13 group and MR-untreated IL-4/IL-13 group. In MR-treated IL-4/IL-13, the therapeutic effects of ruxolitinib and anti-IL4Rα were substantially diminished compared to the MR-untreated IL-4/IL-13. Between MR-treated and MR-untreated groups, it is clear that MR generally reduces the expression levels of CerS3 and ELOVL1 (Appendix A).

### 2.3. Epidermal Permeability Barrier of the AD-RHE Stimulated with Malassezia Restricta (MR)

To investigate the epidermal permeability barrier of the RHE, we imaged the intercellular lipid layer (ILL), corneodesmosome, and lamellar bodies (LBs) using TEM. Normally, ILLs are located within the SC and are shaped like straight hoses. They are filled with lipid and corneodesmosomes, and several LBs are attached under the ILL. Unfortunately, LBs were only seen in the control group but not in MR-treated or IL-4/IL-13-treated RHE (Figure 3 and Appendix A).

In the MR-untreated IL-4/IL-13 group, the ILL appeared thick and wavy, and the intercellular lipid was decreased. Anti-IL4Rα or ruxolitinib treatment restored the curvature of the ILL, resulting in more lipid filling inside the ILL compared to the IL-4/IL-13-only treatment group (Figure 3B–D).

In the MR-treated groups, the ILL appears thicker in the MR-only treated RHE compared to the control RHE (Figure 3E). After additional IL-4/IL-13 treatment, the ILL thickened even further, and wrinkles appeared. Even though the thickness of the ILL was greater than that of the MR-untreated group, the ILL contained less lipid filling. In addition, the corneodesmosomes had lower densities. When the MR-treated IL4/IL13 was further treated with anti-IL4Rα and ruxolitinib, the morphology of ILL was partially recovered, but that of the corneodesmosomes was not (Figure 3E–H).

### 2.4. Expression of Genes Related to Th1, Th2, Th17 Inflammation in AD-RHE Stimulated with Malassezia Restricta (MR)

We examined the expression of Th2 inflammatory markers and vascular endothelial growth factor (VEGF) at the mRNA levels. As shown in Figure 4, IL-4, IL-4Rα, CCL17, CCL22, TSLP, and VEGF are all noticeably upregulated by IL-4/IL-13 or MR treatment; higher expressions are shown in the MR-treated IL-4/IL-13 groups compared to the control group (Figure 4A–F). In both the MR-treated and the MR-untreated groups, anti-IL4Rα and ruxolitinib downregulated Th2-related gene expression increased by IL-4/IL-13 treatment. Interestingly, in TSLP and VEGF, anti-IL4Rα treatment in the MR-treated IL-4/IL-13 group showed significant restoration compared to the MR-only group, while ruxolitinib treatment showed no significance. Between those groups, it is clear that MR generally raises the expression level of Th2-related genes (Appendix A). However, the only statistical significance found between the MR-untreated IL-4/IL-13 and the MR-treated IL-4/IL-13 groups was shown in VEGF, not in Th2 cytokines. The therapeutic effect of anti-IL4Rα and ruxolitinib was lower in the MR-treated groups compared to the MR-untreated groups (Appendix A). 

We also examined the expression of Th17-related genes and Th1-related genes (IFN-γ, CCL20) at the mRNA level (Figure 4G–L). IL-4/IL-13 or MR treatments significantly increased the expression level of Th17-related genes compared to the control (Figure 4F–I); the expression level was even greater in MR-treated IL-4/IL-13 compared to the control. Th17-related gene (IL-1α, IL-17, IL-22, TNF-α) expression was significantly suppressed by anti-IL4Rα in both the MR-treated and the MR-untreated groups. However, the inhibitory effect of ruxolitinib was not significant in IL-22 in the MR-treated group. Between MR-treated and MR-untreated groups, it is evident that MR generally increases the level of expression of Th17-related genes (Appendix A). In particular, the expression level of TNF-α increased significantly in the MR-treated IL-4/IL-13 group compared to the MR-untreated group (Appendix A). The expression level of CCL20 was also significantly increased in the MR-treated IL-4/IL-13 group compared to the control (Figure 4L). IFN-γ exhibited no remarkable variations (Figure 4K and Appendix A). The degree of restoration in the MR-treated groups was generally lower than that in the MR-untreated groups in Th17-related genes.

### 2.5. Protein Expression of Th2- and Th17-Related Molecules in AD-RHE Stimulated with Malassezia Restricta (MR)

We investigated the Th2- and Th17-related molecules at the protein level by Western blotting (Figure 5). The protein levels of Th2-related molecules (TSLP and IL-4) were increased by IL-4/IL-13 or MR treatments; higher expressions were shown in the MR-treated IL-4/IL-13 group compared to the control group. In both the MR-untreated IL-4/IL-13 and the MR-treated IL-4/IL-13 groups, TSLP and IL-4 expressions were decreased by further anti-IL4Rα and ruxolitinib treatments, similar to their mRNA levels (Appendix A).

Th17-related molecules (IL-1β, IL-17) were also upregulated by IL-4/IL-13 or MR treatment; higher expressions were shown in the MR-treated IL-4/IL-13 group compared to the control group. Between MR-treated and MR-untreated groups, it is evident that MR generally increases the protein level of IL-17 (Appendix A). Overall, the level of Th17 inflammatory markers was reversed by ruxolitinib and anti-IL4Rα, but MR-treated groups did not show significant recovery compared to the MR-untreated groups. The protein expression level of IL-22 in the MR-treated IL-4/IL-13 group did not show a significant increase; however, a significant increase was shown at the transcriptional level (Figure 5B).

### 2.6. Protein Expression of JAK/STAT pathway-Related Molecules in AD-RHE Stimulated with Malassezia Restricta (MR)

We examined the JAK/STAT pathway-related molecules at the protein level using Western blotting (Figure 6). Overall, the expressions of total and phosphorylated JAK1, JAK2, and STAT3 were upregulated by IL-4/IL-13 treatment compared to the control. The phosphorylation levels of JAK1, JAK2, and STAT3 in MR-untreated AD-RHE were considerably suppressed by ruxolitinib and anti-IL4Rα. The expression of JAK1 and JAK2 was significantly increased in MR-treated AD-RHE compared to the MR-only treated RHE. In the MR-treated AD-RHE, the phosphorylation of JAK1 and JAK2 levels were suppressed by ruxolitinib and anti-IL4Rα. However, the therapeutic effect of anti-IL4Rα and ruxolitinib was lower in the MR-treated groups compared to the MR-untreated groups. Interestingly, the levels of JAK1, JAK2, and STAT3 phosphorylation that ruxolitinib suppressed in RHE were all comparable to those suppressed by anti-IL4Rα. Comparing the MR-treated and MR-untreated groups, MR treatment upregulated relative fold changes of JAK2 and STAT3 (Appendix A).

## 3. Discussion

In this study, we sought to examine the interactions between *M. restricta*, ruxolitinib, and anti-IL4Rα in the AD model in 3D skin. We established healthy skin-like RHE, AD-mimetic RHE, and AD lesion ongoing treatment-like RHE, proven by histological changes, suppressed skin barrier molecules, and Th2-deviated cytokine expression. Both ruxolitinib and anti-IL4Rα effectively improved skin barrier functions and Th2-deviated inflammatory responses in IL-4/IL-13-treated RHE when *Malassezia* was absent. Our study demonstrated that MR overgrowth damaged skin barrier function by affecting ceramide, LOR, and morphology of the LB and promoted Th2 and Th17 cytokine mRNA transcription even in healthy RHE. MR treatment exacerbated Th2 and Th17 inflammation after IL-4/IL-13 treatment in the AD-RHE model. However, the treatments with anti-IL4Rα or ruxolitinib showed less efficacy in both barrier function recovery and anti-inflammatory effects when MR is present on the surface of RHE. 

Histologically, AD lesional skin shows hyperkeratosis, acanthosis, and spongiosis [36]. Studies have suggested that activating the JAK/STAT signaling pathway and producing Th2 cytokines may also contribute to developing spongiosis in AD [22,24,36]. This was also confirmed in our research results (Figure 1 and Figure 6). The increased spongiosis and epidermal thickness demonstrate that AD modeling was successfully created in our study. Spongiosis, acanthosis, and the exfoliation of the SC layer induced by MR treatment in AD-RHE may explain why head and neck dermatitis in AD patients present lichenification and scales in the *Malassezia*-rich area [18,37]. This may result because the compound secreted by MR or metabolites can exacerbate AD by stimulating cytokine and chemokine secretion.

Epidermal barrier dysfunction is significant in AD because it increases the risk of being affected by environmental triggers [38,39]. As shown in Figure 2, FLG, LOR, and IVL expressions were significantly reduced after IL-4/IL-13 treatment in both MR-untreated and MR-treated groups. However, the MR treatment only did not affect FLG and IVL. Lipid synthase CerS3 expression was decreased in the presence of MR and exacerbated by an additional IL-4/IL-13 treatment (Figure 2). Disruption of either lipid lamellae or corneodesmosome may impair the SC barrier function, resulting in increased permeability, dryness, and inflammation [40,41]. In the control RHE, our TEM result showed a healthy tissue morphology similar to the 3D skin TEM results of other studies [42]. Our TEM results showed that lipid lamellae and corneodesmosome formation were inhibited in MR-treated groups and exacerbated by an additional IL-4/IL-13 treatment (Figure 3). Therefore, MR has a negative impact on the lipid barrier when it exists as a high colony, which can be more harmful in the AD environment. In addition, ruxolitinib or anti-IL4Rα treatments help strengthen the barrier by improving the AD environment, but the effect may not be complete in *Malassezia* overgrowth situations.

AD patients are known to have decreased levels of LL-37 in the skin [43]. While β-defensin2 levels are slightly increased in AD compared to healthy individuals, the levels are amplified in psoriasis [44,45]. β-defensin2 and LL-37 stimulate T-cells to secrete inflammatory cytokines such as IL-4, IL-13, and IL-31 to regulate the itch sensation and increase vascular permeability [45]. We previously showed that *Malassezia* increases β-defensin2 production in human keratinocytes [14]. In our study, the MR-treated groups had significantly higher levels of β-defensin2 and LL-37 compared to the MR-untreated groups (Figure 2 and Appendix A). This suggests that *Malassezia*-induced excessive AMP production may have detrimental effects on AD patients’ inflammatory conditions. AMPs also play an important role in the maintenance of microbiome balance [46]. AMPs secreted by exposure to the normal flora of *Malassezia* can help reduce harmful bacteria such as *S. aureus* and maintain balance [47]. According to recent studies, imbalance of AMP production may lead to changes in the microbiome balance and, consequently, increased inflammation [48,49]. Our results demonstrate that overproduction of AMPs by MR overgrowth can exacerbate AD.

We investigated the impact of MR on the AD environment from an inflammatory point of view. MR overgrowth was enough to increase the expression level of Th2-related genes. Similar to AD lesions, Th2 cytokines were increased in our AD-RHE model, and MR further increased their expressions (Figure 4 and Figure 5) at the mRNA and protein levels. Interestingly, VEGF was also highly increased in the AD-RHE model, and MR further upregulated its expression (Figure 4). VEGF can also modulate immune response by promoting the expression of inflammatory cytokines and chemokines in keratinocytes. Increased VEGF could promote the development of erythema by angiogenesis and the influx of inflammatory cells [50]. These results support the conclusion that *Malassezia* can accelerate AD by accelerating inflammation through the secretion of VEGF and Th2 cytokines. Prior to the experiment, we predicted that the increase in VEGF would be responsible for the redness brought on by the use of dupilumab in a subset of AD patients [31,32]. However, in the experiment, when anti-IL4Rα treatment was applied to the MR-treated AD group, the expression level of VEGF was significantly reduced compared to the MR-only group. These findings suggest that dupilumab-related facial redness is not directly caused by the drug, but it could be related to *Malassezia* overgrowth inducing Th2 inflammation and VEGF expression. Further studies are needed to reveal the therapeutic mechanisms of ruxolitinib in treating dupilumab-related facial redness.

Th1 cells are activated in response to chronic skin inflammation, producing IFN-γ and other Th1 cytokines that contribute to the tissue damage and chronic inflammation seen in AD [51]. Other studies suggest that Th1 cell immunity may be inhibited in acute AD since Th2 cytokines can inhibit the production of IFN-γ and other Th1 cytokines [52]. In our study, both Th1-related cytokines, IFN-γ and CCL20, were not significantly changed in the AD-RHE model; CCL20 was slightly increased only in the MR-treated AD group (Figure 2). This is consistent with the finding that *Malassezia* metabolites augmented the expression of CCL20 mRNA in human epidermal keratinocytes [14].

IL-17 activates macrophages expressing TNF-α and IL-1β, and IL-22 induces epidermal proliferation [53]. Similarly, in our AD-RHE model, IL-17 and IL-22 increased at both the mRNA and protein levels, and IL-1α and TNF-α increased at the mRNA level. The presence of MR further increased the expression of IL-1α, TNF-α, and IL-17 mRNA (Figure 4 and Figure 5). Exposure to *Malassezia* produces cytokines related to IL-17 to defend the skin from fungal infection [54]. *Malassezia* increases skin inflammation caused by IL-23 and IL-17 in skin barrier dysfunction, which mimics AD [55]. It is reported that AD skin had a higher Th17 subset of memory T cells specific to *Malassezia* than healthy skin. Our study shows that the expression of IL-17 protein was increased by MR treatment; the MR-treated groups showed significantly higher expression levels than MR-untreated groups when treated with ruxolitinib or anti-IL4Rα in the AD-RHE (Figure 4 and Figure 5). This may result from antifungal immunity being restored by the ruxolitinib or anti-IL4Rα, but the therapeutic efficacy of anti-Th17 inflammation was not sufficient in a *Malassezia*-rich environment.

*Malassezia* species form dense colonies to produce Th2 cytokines and chemokines in keratinocytes, resulting in T cell activation and recruitment of other inflammatory cells by upregulating adhesion molecules [56,57,58]. In our study, we inoculated MRs 1–3 × 10^8^/cm^2^ per well. This concentration corresponds to an MR to keratinocyte ratio that equals to 20–30:1 in 2D keratinocyte culture. Thus, the experiment was conducted at an infection-like state induced by MR overgrowth. The level of Th2-related cytokines was significantly upregulated in the MR-treated groups compared to the MR-untreated groups (Appendix A). The results suggest that *Malassezia* overgrowth can exacerbate Th2 inflammation even in healthy skin. Moreover, MR with AD-RHE had a higher expression level of Th2 cytokines. This is consistent with the findings that *Malassezia* exacerbates AD through the secretion of Th2 cytokines in keratinocytes. 

As an extension of inflammation, the effect of MR on the JAK/STAT pathway, the downstream signaling of Th2 cytokines, was also investigated. The activation of the JAK/STAT pathway upregulates TSLP and downregulates FLG, LOR, and IVL expression to disrupt the skin barrier function Both IL-4 and IL-13 bind to IL-4R/IL-13R1 in keratinocytes, activating downstream JAK1/JAK2 and the STAT3/STAT6 pathways. [59,60]. Consistent with our results, phosphorylated and total forms of JAK1, JAK2, and STAT3 were upregulated in IL-4/IL-13-treatment RHE and downregulated after ruxolitinib and anti-IL4Rα treatment (Figure 6). Interestingly, MR-treated groups showed a significant increase in the protein levels of JAK1, JAK2, and STAT3 compared to MR-untreated groups (Appendix A). In conjunction with the expression of TSLP, FLG, LOR, and IVL in Figure 2 and Figure 4, we can deduce that MR activates the JAK/STAT pathway and subsequently regulates the expression of TSLP and skin barrier molecules.

In our study, anti-IL4Rα and ruxolitinib were generally successful in restoring skin barrier and inflammatory responses in the AD-RHE model (Figure 1, Figure 2, Figure 3, Figure 4, Figure 5 and Figure 6, Table 1). In particular, anti-IL4Rα was significantly effective in recovering VEGF, Th2, and Th17-related molecules but partially effective in recovering skin barrier molecules. In the MR-treated AD-RHE, only anti-IL4Rα showed a significant therapeutic effect, but the recovery was less competent than that of the MR-untreated groups. It is known that secreted proteases or metabolites from *Malassezia* could damage the skin barrier [18,19]. We suspect this may result in more barrier defects and reduced ability of new drugs to restore those barrier defects and inflammation in the MR-treated groups. It is anticipated that a more complicated mechanism drives AD exacerbation by MR, and further research is needed on this topic.

The limitation of our study is that 3D skin is only comprised of keratinocyte layers, and devoid of inflammatory cells and blood vessels, which may not accurately reflect the infiltration of inflammatory cells or the secretion of inflammatory substances. To overcome this, collecting AD patients’ skin samples and *Malassezia* colonies before and after treatment with JAK inhibitors or anti-IL4Rα antibodies would be beneficial.

## 4. Materials and Methods

### 4.1. Malassezia Culture

*M. restricta* strains (KCTC 27527) isolated from human scalp was used throughout the study [61]. *M. restricta* strains was grown in Leeming and Notman agar medium (LNA; pH6, 0.5% glucose, 1% polypeptone, 0.01% yeast extract, 0.8% bile salt, 0.1% glycerol, 0.05% glycerol monostearate, 0.05% Tween 60, 1.2% agar, and 0.5% whole fat cow milk) for 3 days at 34 °C [62]. 

After being grown for 3 days at 34 °C, *M. restricta* was resuspended in modified Dixon’s medium (mDixon’s; 3.6% malt extract, 2% bile salt, 1% Tween 40, 0.6% polypeptone, 0.2% oleic acid, and 0.2% glycerol) and inoculated on the keratinocytes or RHE.

### 4.2. HaCaT Cell Culture

We purchased the human immortalized keratinocyte cell line (HaCaT cells) from American Type Culture Collection (ATCC, Manassas, VA, USA). HaCaT cells at passages 30 to 40 were used for the experiments. The cells were cultured in DMEM medium (Gibco, Gaithersburg, MD, USA) supplemented with 10% FBS (Gibco BRL, Life Technology, Karlsruhe, Germany) and 1% penicillin/streptomycin (Gibco, BRL, Life Technology, Karlsruhe, Germany) at 37 °C under 5% CO_2_ conditions. HaCaT cells (1 × 10^5^ cells/well) were seeded into 24-well plates with serum-free DMEM, and the plates were treated with various concentrations of recombinant IL-4/IL-13 (PeproTech, Rocky Hill, NJ, USA) and exposed to *M. restricta* at a yeast cell:keratinocyte ratio of 20–30:1 for 48 h.

### 4.3. Setting Up an In Vitro 2D AD Model

The AD-induced IL-4/IL-13 concentration was selected through a pilot experiment using keratinocytes. An experimental model was set up by using different concentrations of IL-4 and IL-13 (3 ng/mL, 10 ng/mL, 30 ng/mL) for 48 h. Through PCR, it was confirmed that the expression of AD-related genes (IL-4, IL-4Rα) increased significantly when the IL-4/IL-13 dose was 10 ng/mL or greater (Appendix A). In addition, the concentration of IL-4/IL-13 was set through cell viability testing using the MTT assay at 10 ng/mL, which is ideal for cell viability (Appendix A). We used 10 ng/mL of IL-4/IL-13 and 500 nM of ruxolitinib for this study in all experiments. 

### 4.4. 3D Skin AD Modeling

An RHE (Keraskin™, Biosolution Co., Seoul, Korea) was used as an in vitro model of the human epidermis. According to the datasheet, the RHE samples were transferred to 6-well plates with 0.9 mL/well of Keraskin culture medium (Keraskin™, Biosolution Co., Seoul, Korea) for 24 h at 37 °C and 5% CO_2_. After being pre-incubated, the RHE samples were transferred to fresh medium and treated with 60ul of *M. restricta* (1–3 × 10^8^/well) diluted in mDixon for 24 h. After 24 h treatment of *M. restricta*, the AD groups’ RHE were stimulated with recombinant human IL-4 (10ng/mL, Peprotech, Rocky Hill, NJ, USA) and IL-13 (10 ng/mL, Peprotech, Rocky Hill, NJ, USA) to induce an in vitro AD model. Two hours after IL-4/IL-13 treatment, the RHE were treated with anti-IL4Rα (1 µg/mL, Santa Cruz Biotechnology, Inc., CA, USA) or ruxolitinib (500 nM/mL, Invivogen, San Diego, CA, USA) for 48 h to simulate treated AD lesions. An equal volume of vehicles was used simultaneously in the control group. After treatment, the RHE was harvested for analysis. The experiment was repeated three times. 

### 4.5. Cell Viability Assay

MTT assay was used to calculate cell viability. After co-culturing with different concentrations, the medium and *M. restricta* were removed, and the HaCaT cells were washed once with DPBS. Next, MTT reagent (0.5 mg/mL in DMEM) was added and reacted for 3 h at 37 °C incubation. After the reagent was removed, the crystals of formazan that were left were dissolved in dimethylsulfoxide (DMSO). Then, the samples were evaluated by a microplate spectrophotometer (NanoDrop spectrophotometer; Thermo Scientific, Fremont, CA, USA), measuring the absorbance at 540 nm. 

### 4.6. Real-Time PCR

Total RNA was extracted from the RHE using TRIzol reagent (Invitrogen, Carlsbad, CA, USA), and the isolated RNA was measured by spectrophotometer (NanoDrop spectrophotometer; Thermo Scientific, Fremont, CA, USA). After being quantified, cDNA was synthesized using the MG cDNA Synthesis Kit (CancerROP, Seoul, Korea) according to the manufacturer’s recommendations. Real-time PCR was performed using this cDNA with SYBR Green Master mix (CancerROP, Seoul, Korea) by CFX96 Touch Real-Time PCR (Bio-Rad, Hercules, CA, USA). The gene expression levels were quantified using analysis software (CFX Manage Software 3.1, Bio-Rad, Hercules, CA, USA). The primers were created and purchased from Bioneer (Bioneer, Daejeon, Korea); the primer sequences are listed in Appendix A. GAPDH (glyceraldehyde-3-phosphate dehydrogenase) was used as a housekeeping gene, and values were computed using the delta-delta CT method. 

### 4.7. Western Blot Analysis

RHE samples were harvested using RIPA lysis buffer added with Protease and Phosphatase Inhibitor Cocktail (ThermoFisher, Rockford, IL, USA) for protein extraction. The protein concentration was calculated using the BCA protein assay kit (ThermoFisher, Rockford, IL, USA), and the results were compared to BSA standards (ThermoFisher, Rockford, IL, USA). Samples with the same amount of total protein were separated using electrophoresis on sodium dodecyl sulfate (SDS)-polyacrylamide gels, and the samples were then transferred to polyvinylidene fluoride (PVDF) membranes using iBlot (ThermoFisher, Rockford, IL, USA). The membranes were blocked with 5% BSA in TBST for 1 hour at room temperature (RT) and were probed with primary antibodies in 5% BSA in TBST at 4 °C overnight. Primary antibodies against phospho-JAK1-2, total-JAK1-2, phospho-STAT3, total-STAT3, IL-1β, IL-4, IL-17, IL-22, Thymic Stromal Lymphopoietin (TSLP), Filaggrin (FLG), Loricrin (LOR), Involucrin (IVL), LL-37, β-defensin2, and β-actin were purchased from Santa Cruz Biotechnology (Santa Cruz, CA, USA) and Cell Signaling Technology, Inc. (Cell Signal, Beverly, MA, USA). After several TBST washes, the membranes were incubated for 2 h at RT with peroxidase-conjugated secondary antibodies (Cell Signal, Beverly, MA, USA). Proteins were detected using ECL detection system (ThermoFisher, Rockford, IL, USA) and were photographed using a chemiluminescence imaging system (ChemiDoc Imaging System; Biorad, Hercules, CA, USA). 

### 4.8. H&E Staining and Immunofluorescent Staining

For the histological examination, the RHE samples were fixed in 4% paraformaldehyde (Tech and Innovation, Chuncheon-si, Gangwon-do, Korea), submerged in sucrose (Sigma-Aldrich, Hamburg, Germany), and embedded in a cryomold using O.C.T. compound (Sakura Finetek, CA, USA). The embedded RHE samples were cut into 8µm thickness by cryo-microtome and attached to slide glass. The RHE samples were incubated in a SuperBlock blocking solution (ThermoFisher, Rockford, IL, USA) for 30 min at RT to prevent nonspecific staining.

For the H&E staining, the RHE samples were stained with hematoxylin solution (Sigma-Aldrich, St. Louis, MO, USA) and washed with tap water; when the color was ideal, the samples were stained with eosin solution (Sigma-Aldrich, St. Louis, MO, USA). Next, the stained samples were dehydrated with EtOH and cleared with xylene (DUKSAN, Seoul, Korea). Then, samples were covered with a Dako mounting solution (Dako, Baar, Switzerland) and observed using a light microscope (Olympus, Tokyo, Japan). 

For the immunofluorescent staining, the RHE samples were immunostained with an anti-Filaggrin antibody (1:200, Abcam, Fremont, CA, USA), anti-Loricrin antibody (1:200, Abcam, Fremont, CA, USA), and anti-Involucrin antibody (1:200, Abcam, Fremont, CA, USA) overnight at 4 °C. The following day, samples were incubated for 1 hour at RT with an Alexa Fluor-488 (1:400, Abcam, Fremont, CA, USA) conjugated secondary antibody and then covered with a Vector Shield mounting medium with DAPI (Vector Laboratories, Newark, CA, USA). The stained samples were examined with a fluorescence microscope (Axiovert 200, Zeiss, Oberkochen, Germany) at 20× magnification. Three randomly chosen photographs from each RHE sample were used to quantify. 

### 4.9. Transmission Electron Microscopy (TEM)

The RHE samples were incubated in 2.5% Glutaraldehyde-Paraformaldehyde (Sigma-Aldrich, Hamburg, Germany) for 24 h and washed in 0.1 M phosphate buffer (PB) (phosphate buffer). Then, samples were fixed with 1% osmium tetroxide (OsO4) (phosphate buffer) dissolved in 0.1 M PB for 1 h and washed in distilled water. The samples were then dehydrated with ethanol stepwise and infiltrated with propylene oxide (phosphate buffer). Specimens were embedded by epoxy resin (Epon 812 resin, EMS, Hatfield, PA, USA) and polymerization at 60 °C in a vacuum oven (ThermoFisher, Rockford, IL, USA, DOSASKA, 3618-1CE) for 24 h. Sections were transferred to copper and nickel grids after being cut in 100nm with a diamond knife by the LEICA EM UC-7 (Leica Microsystems, Wetzlar, Germany). Samples were examined under transmission electron microscopy (JEM-1010, JEOL, Tokyo, Japan) at a voltage of 60 kV. TEM imaging and organization of the resulting photographs were conducted with reference to the cited literature [63].

### 4.10. Statistical Analysis

All data are expressed as mean ± SEM. A one-way ANOVA was used to compare three groups, and the Student’s *t*-test was used to compare two groups. A *p*-value less than 0.05 was regarded as statistically significant. Data were statistically analyzed using GraphPad Prism software (San Diego, CA, USA, Prism Windows 7.05).

## 5. Conclusions

This is a new report on the role of MR in AD-RHE and the effect of ruxolitinib or anti-IL4Rα on skin barrier function and inflammation. We confirmed that MR overgrowth could affect the skin barrier by increasing AMPs, deteriorating the lipid barrier, and stimulating the secretion of Th2 cytokines. Furthermore, in the AD microenvironment, MR amplified the secretion of AMPs, Th2, and Th17 cytokines, which activated the JAK/STAT signaling pathway and further reduced the production of skin barrier proteins, causing lipid barrier abnormalities. Additional treatment with anti-IL4Rα downregulated the levels of Th2- and Th17-inflammatory markers, and both anti-IL4Rα and ruxolitinib exhibited limited restoration of skin and lipid barrier in the presence of MR. Our findings suggest that MR exacerbates AD in both the skin barrier and inflammation and decreases the effectiveness of treatment with ruxolitinib and dupilumab.

## Figures and Tables

**Figure 1 ijms-24-06171-f001:**
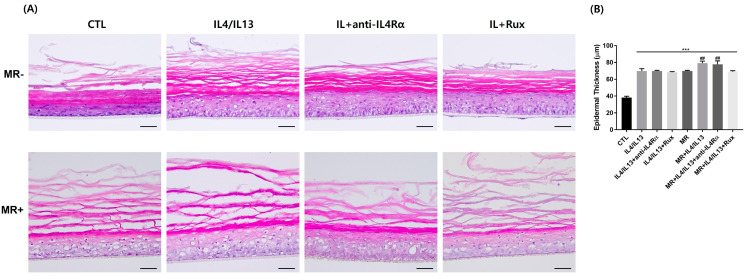
Changes in tissue morphology of the AD-RHE stimulated with *Malassezia Restricta* (MR). (**A**) H&E staining of RHE. IL-4/IL-13 treatment causes hyperkeratosis, acanthosis, spongiosis. MR treatment also causes spongiosis and SC deformation; those effects were amplified in the MR-treated IL-4/IL-13 RHE. (**B**) Epidermal thickness of RHE. IL-4/IL-13 or MR treatment caused an increase in epidermal thickness. Error bars represent the mean ± SEM, n = 3. Statistically significant at *** *p* < 0.001 compared to the control (CTL), and ## *p* < 0.01 compared to the MR-only treated RHE (MR). Scale bar = 50 µm.

**Figure 2 ijms-24-06171-f002:**
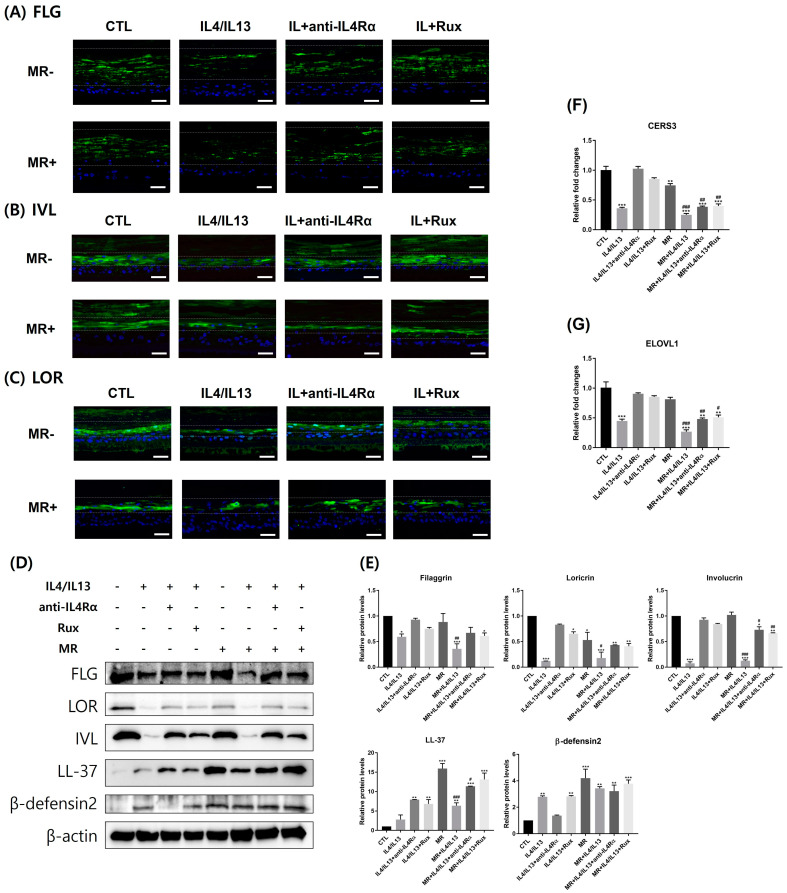
Analysis of epidermal skin barrier and antimicrobial peptide-related molecules by immunofluorescence, Western blotting, and PCR. (**A**) FLG, (**B**) IVL, and (**C**) LOR expressions are shown in green. Nuclear staining was performed using DAPI (blue). In MR-untreated groups, the expression levels of skin barrier molecules decreased with IL-4/IL-13 treatment and were restored with anti-IL4Rα or ruxolitinib treatments. In MR-treated groups, a decreased recovery with anti-IL4Rα and ruxolitinib treatment was shown. (**D**) Western blotting band images and (**E**) quantification graph of Western blotting bands. Expression levels of the markers were evaluated by Western blotting and β-actin was used as a loading control. The levels of FLG, IVL, and LOR were downregulated in MR-untreated IL-4/IL-13 groups and in MR-treated IL-4/IL-13 groups, and upregulated by anti-IL4Rα and ruxolitinib treatments. LL-37 and β-defensin2 were upregulated by MR treatment. The mRNA levels of (**F**) CerS3 and (**G**) ELOVL1 were downregulated by IL-4/IL-13 or MR and reverted by anti-IL4Rα or ruxolitinib. Error bars represent the mean ± SEM, n = 3. Statistically significant at * *p* < 0.05, ** *p* < 0.01, and *** *p* < 0.001 compared to the control (CTL) and # *p* < 0.05, ## *p* < 0.01, and ### *p* < 0.001 compared to the MR-only treated RHE (MR). Scale bar = 50 µm.

**Figure 3 ijms-24-06171-f003:**
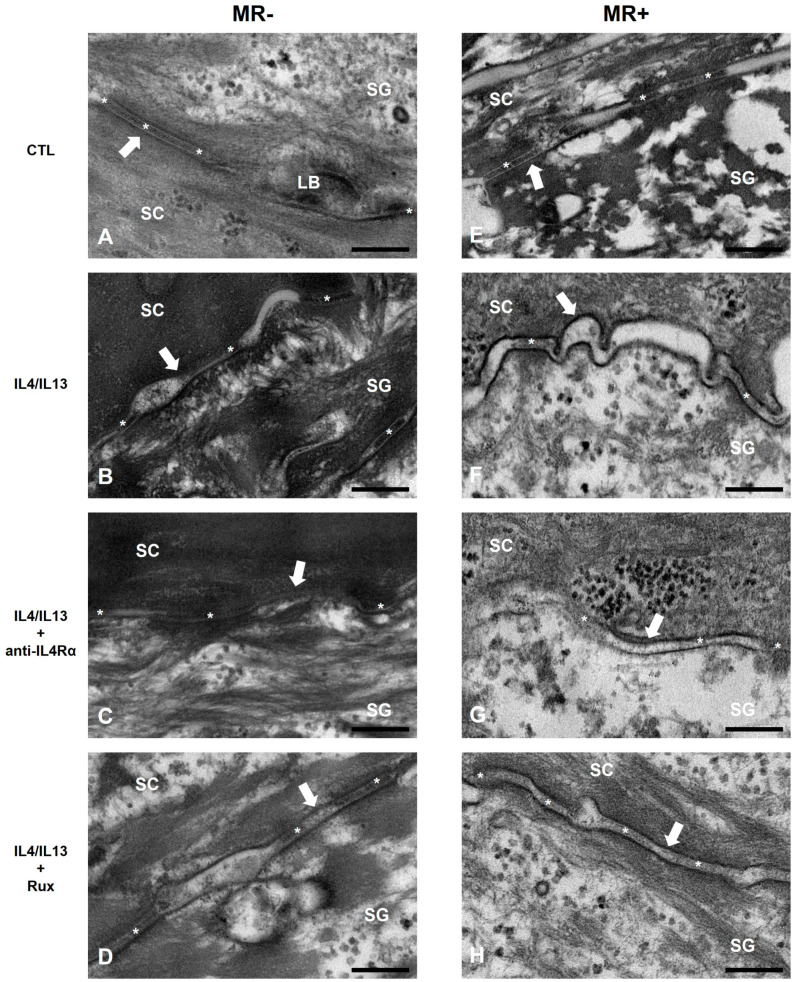
TEM images of the intercellular lipid layer and corneodesmosome in the intercellular space. (**A**) CTL, (**B**) IL-4/IL-13, (**C**) IL-4/IL-13/anti-IL4Rα, (**D**) IL-4/IL-13/ruxolitinib, (**E**) MR, (**F**) MR-treated IL-4/IL-13, (**G**) MR-treated IL-4/IL-13/anti-IL4Rα, and (**H**) MR-treated IL-4/IL-13/ruxolitinib. In MR-untreated groups, ILL thickened when treated with IL-4/IL-13 and recovered when treated with anti-IL4Rα or ruxolitinib. In the MR-treated group, ILL elongated and curved when treated with IL-4/IL-13, and the density of corneodesmosome was lowered. After anti-IL4Rα or ruxolitinib treatment, the contour of ILL was slightly recovered, but the morphological alternations of the corneodesmosome was not significantly recovered. SC: stratum corneum, SG: stratum granulosum, LB: lamellae body, white arrow: intercellular lipid layer (ILL), asterisk: corneodesmosome. Scale bar = 0.2 µm.

**Figure 4 ijms-24-06171-f004:**
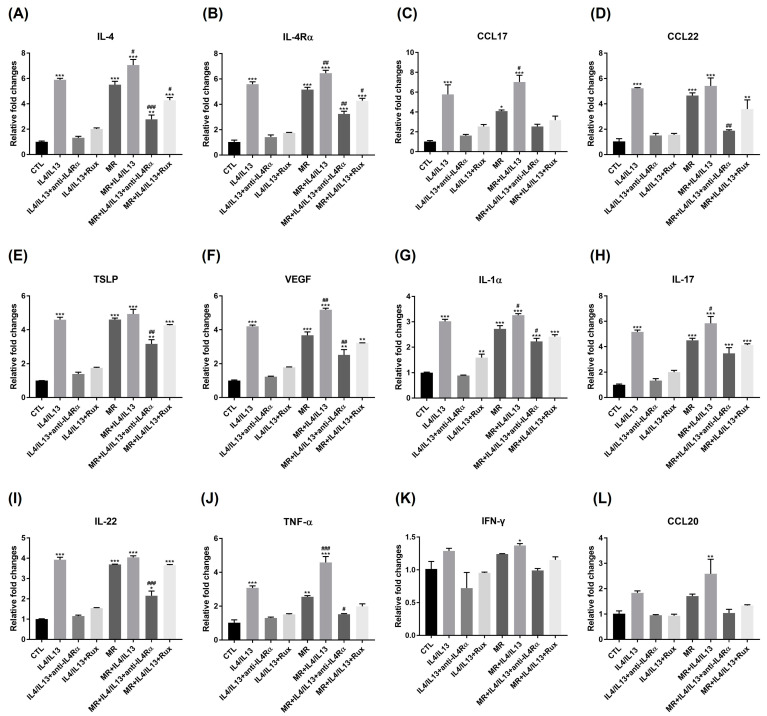
Analysis of Th1-, Th2-, and Th17-related genes expressions by RT-PCR. The mRNA level of (**A**) IL-4, (**B**) IL-4Rα, (**C**) CCL17, (**D**) CCL-22, (**E**) TSLP, (**F**) VEGF, (**G**) IL-1α, (**H**) IL-17, (**I**) IL-22, (**J**) TNF-α, (**K**) IFN-γ, and (**L**) CCL20 in RHE. Th2-related genes and VEGF (**A**–**F**) were upregulated by IL-4/IL-13 or MR and reverted by anti-IL4Rα or ruxolitinib. Th17-related genes (**G**–**J**) were upregulated by IL-4/IL-13 or MR and downregulated by anti-IL4Rα or ruxolitinib. Th1-related genes (**K**,**L**) only showed a significant increase in MR-treated IL-4/IL-13. Error bars represent the mean ± SEM, n = 3. Statistically significant at * *p* < 0.05, ** *p* < 0.01, and *** *p* < 0.001 compared to the control (CTL) and # *p* < 0.05, ## *p* < 0.01, and ### *p* < 0.001 compared to the MR-only treated RHE (MR).

**Figure 5 ijms-24-06171-f005:**
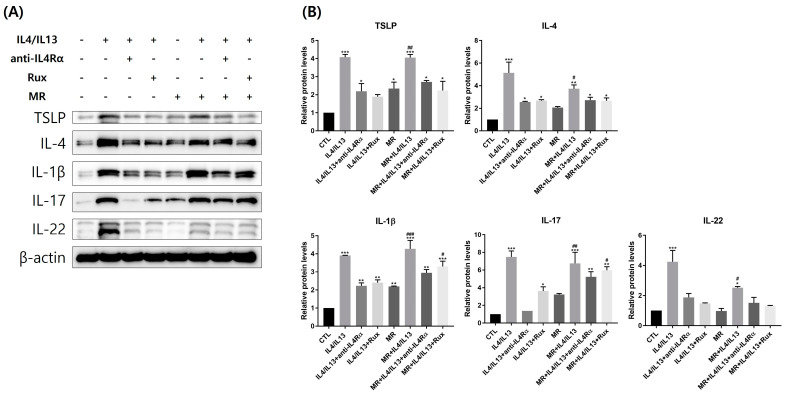
Analysis of Th2- and Th17-related molecules’ protein levels in AD-RHE stimulated with *Malassezia Restricta* (MR). (**A**) Western blotting band images and (**B**) quantification graph of Western blotting bands. Expression levels of the markers were evaluated by Western blotting and β-actin was used as a loading control. Th2 and Th17 markers were upregulated by IL-4/IL-13, MR, and MR-treated IL-4/IL-13 groups and suppressed by anti-IL4Rα or ruxolitinib. IL-22 was increased in MR-untreated IL-4/IL-13 and M-treated IL-4/IL-13 but not in MR-only treated RHE. Error bars represent the mean ± SEM, n = 3. Statistically significant at * *p* < 0.05, ** *p* < 0.01, and *** *p* < 0.001 compared to the control (CTL), and # *p* < 0.05, ## *p* < 0.01, and ##*#* < 0.001 compared to the MR-only treated RHE (MR).

**Figure 6 ijms-24-06171-f006:**
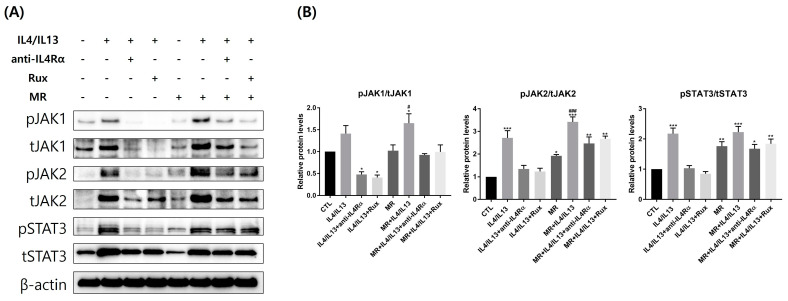
Changes in JAK/STAT pathway-related molecules in AD-RHE stimulated with *Malassezia Restricta* (MR). (**A**) Western blotting band images and (**B**) relative fold changes show that MR-untreated IL-4/IL-13 or MR-treated IL-4/IL-13 groups increase the levels of phosphorylated JAK1, JAK2, and STAT3. Activated signaling of the JAK/STAT pathway by IL-4/IL-13 was remarkably suppressed by ruxolitinib or anti-IL4Rα. Error bars represent the mean ± SEM, n = 3. Statistically significant at * *p* < 0.05, ** *p* < 0.01, and *** *p* < 0.001 compared to the control (CTL), and # *p* < 0.05 and ### *p* < 0.001 compared to the MR-only treated RHE (MR).

**Table 1 ijms-24-06171-t001:** Summary table of the study.

	Normal Skin IL-4/IL-13 (-)		Barrier Disrupted Skin IL-4/IL-13 (+)		Barrier Recovered Skin IL-4/IL-13 (+) with anti-IL4Rα or Ruxolitinib	
With *M. restricta* vs. Without *M. restricta*	MR (-)	MR (+)	MR (-) ^#^	MR (+)	MR (-) ^†^	MR (+)
Skin barrier molecules	FLG, LOR, IVL	LOR↓	FLG↓ LOR, IVL↓↓	↓ *	FLG, LOR↑ IVL↑↑	LOR, IVL↓
Lipid synthesis	CERS3, ELOVL1	CERS3↓	↓↓	↓ *	↑↑	CERS3↓↓, ELOVL1↓
Vascular	VEGF	↑	↑↑	↑	↓↓	↑
Innate immunity	IL-1β IL-17, IL-22 LL-37, β-D2	IL-1β↑ IL-17↑ LL-37, β-D2↑↑	IL-1β↑↑ IL-17, IL-22↑↑ β-D2↑	IL-1β↑ * IL-17↑ * LL-37, β-D2↑ *	IL-1β↓ IL-17, IL-22↓↓ LL-37↑	IL-1β↑ IL-17↑ LL-37, β-D2↑
Proinflammatory cytokines and chemokines	IL-4, TSLP IL-1α, TNF-α CCL17, CCL22	IL-4, TSLP↑↑ IL-1α↑↑, TNF-α↑ CCL17↑, CCL22↑↑	↑↑	IL-4, TSLP↑ * CCL17, CCL22↑ * TNF-α↑	IL-4, TSLP↓↓ IL-1α↓↓, TNF-α↓ CCL17↓, CCL22↓↓	IL-4↑, TSLP↑↑
JAK/STAT pathway	JAK1, JAK2, STAT3	JAK2, STAT3↑	JAK2, STAT3↑↑	↑ *	JAK1, JAK2↓ STAT3↓↓	↑

In the MR (-) barrier-disrupted skin, the levels of skin barrier molecules and inflammatory markers were compared to MR (-) normal skin (#). MR (-) barrier-recovered skin molecules were compared to those of MR (-) barrier-disrupted skin (†). The MR (+) groups were compared with MR (-) groups within normal, barrier-disrupted, and barrier-recovered skin. The asterisk (*) signifies the molecular levels of barrier-disrupted RHE were not aggravated by MR treatment compared to those of MR-untreated/barrier-disrupted RHE. Two arrows (↑↑, ↓↓) signify statistical significance of *p* < 0.001, and one arrow (↑,↓) signifies statistical significance of 0.001 < *p* < 0.05.

## Data Availability

The data that support the findings of this study are available from the corresponding author upon reasonable request.

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
