# Peer review of "Interactions between Malassezia and New Therapeutic Agents in Atopic Dermatitis Affecting Skin Barrier and Inflammation in Recombinant Human Epidermis Model"

_ijms, 2023, doi:10.3390/ijms24076171_

Round 1

Reviewer 1 Report

This new research investigates the significance of MR in AD-RHE and the impact of ruxolitinib or anti-IL4Rα on skin barrier function and inflammation. The quality of study is good and can be published in the International Journal of Molecular Sciences. However, there are a few minor corrections after which the article can be published. Here are the following corrections:

1)      The journal’s requirements is that the abstract must be a maximum of 200 words. However, the total number of words is nearly 400, double the maximum. The authors must be reduce the number of words.

2)      The figure 7 should be a table in the format of publication.

3)      The important limitation of this study is that the 3D skin is only comprised of keratinocyte layers and devoid of inflammatory cells and blood vessels. How does the author plan to overcome these limitations in the future?

4)      Also, the authors must cite certain previous articles which has used TEM for similar research and here are some of the examples:

a.      Arvind Mukundan et al.,” Optical and Material Characteristics of MoS2/Cu2O Sensor for Detection of Lung Cancer Cell Types in Hydroplegia,” Int. J. Mol. Sci. 2022, 23(9), 4745 (2022).

b.      Arvind Mukundan et al ,” Growth Mechanism of Periodic Periodic-Structure MoS2 by Transmission Electron Microscopy,” Nanomaterials 2022, 12, 135 (2022).

Author Response

We appreciate your thorough review of this manuscript and hope that we have addressed your concerns, as detailed below.

Point 1: The journal’s requirement is that the abstract must be a maximum of 200 words. However, the total number of words is nearly 400, double the maximum. The authors must be reduce the number of words.

Response 1: Thank you for the comments. We reduced the words of the abstract to under 200.

Point 2: The figure 7 should be a table in the format of publication.

Response 2: Thank you for the delicate insight. We changed the figure 7 to the table format.

Point 3: The important limitation of this study is that the 3D skin is only comprised of keratinocyte layers and devoid of inflammatory cells and blood vessels. How does the author plan to overcome these limitations in the future?

Response 3: We agree with the reviewer’s opinion. An ideal skin model should mimic the actual skin structure and function, including various cells, appendages, fat cells, and blood vessels. However, currently developed skin models have limitations regarding the presence of necessary immune cells, blood vessel maintenance, and cell viability. To overcome this, we plan to collect AD patients’ skin samples before and after treatment with JAK inhibitors or anti-IL4 receptor antibodies. Tape stripping, skin biopsy samples, measurement of transepidermal water loss, and Malassezia colony evaluation would be included in our next study.

Point 4: Also, the authors must cite certain previous articles which has used TEM for similar research and here are some of the examples:

Response 4: We appreciate the valuable comments. According to the reviewer's opinion, additional references related to TEM are described in the discussion.

Reviewer 2 Report

Introduction: I am not sure that the word ''species'' after ''Malassezia'' should be in italic.

Please introduce more the new therapeutic agents that you mentioned in the title. For me was a little difficult to understand the improtance of this study.

Results

2.1) Why the treatment in the the skin with Malassezia did not should any improvement? Are the authors convinced that we have a problem with the colonization by Malasseia that can be different from the approach of these new drugs? Please clarify.

2.2) In figure 2 is possible to observe that IL4 and 13 affects the production of IVL and Malassezia did not. In this sense, the utilization of this drug is once more not pointing for an interesting results regarding Malassezia.  The same I observed for LOR. Can the authors clarify why this drug should be important in the context of Malassezia?

In the conclusions the authors have shown that Malassezia can reduce the skin barrier function and the treatment with this medicament is not effective or that the Malassezia infection can worse during the treatment? Please clarify this.

According to the aims, the authors have declaired: In this study, we investigate how M. restricta affects the AD model in 3D RHE and the effectiveness of ruxolitinib and anti-IL4Rα with respect to the skin barrier function and inflammation.

I am confused regarding the relation of inflamamtion, Malassezia and the purposed drug. Please better clarify this in the present work. What is the central message addressed from your paper?

Figure 7 should not be a figure but a table.

The authors contributions are unclear.

Author Response

We appreciate your thorough review of this manuscript and hope that we have addressed your concerns, as detailed below.

Point 1: Introduction: I am not sure that the word ''species'' after ''Malassezia'' should be in italic.

Response 1: Thank you for the comments. We changed the word “species” to non-italic.

Point 2: Please introduce more the new therapeutic agents that you mentioned in the title. For me was a little difficult to understand the improtance of this study.

Response 2: We appreciate the valuable comments. Ruxolitinib (Jakafi®) and Dupilumab (Dupixent®) are the first topical JAK inhibitors and biologics, respectively, approved by the FDA for treating AD and are currently widely used in the clinic. Some patients treated with dupilumab experience facial redness. Malassezia hypersensitivity has been suggested as one of the hypotheses to explain this phenomenon; however, whether Malassezia is the cause or result of AD flare-ups, whether these adverse effects are a result of interactions between the drugs and Malassezia, or how the drugs affect the skin in the presence of Malassezia are all understudied topics.

Meanwhile, facial redness is not reported in AD patients treated with ruxolitinib. Interestingly,  there are sporadic case reports that JAK inhibitor improved dupilumab-related facial redness. Nevertheless, the therapeutic mechanism of JAK inhibitors is still unknown. We designed normal skin, AD-like skin (IL-4/IL-13 treated RHE), and treated AD lesions-like skin (anti-IL4Rα or ruxolitinib-treated IL-4/IL-13 RHE) using a 3D skin model and examined the role of Malassezia and the drugs in each skin model. This is added in the introduction section.

Point 3: 2.1) Why the treatment in the skin with Malassezia did not should any improvement? Are the authors convinced that we have a problem with the colonization by Malasseia that can be different from the approach of these new drugs? Please clarify.

Response 3: We appreciate the reviewer’s thoughtful comments. To examine the effects of Malassezia, we compared all variables according to the presence of Malassezia colonization (These data are shown in the supplementary figures.). Most Th2 inflammatory markers and skin barrier molecules showed significant differences depending on the existence of Malassezia. Significant changes were also seen in the RHE model, which mimics treated AD lesions using the new drugs. Although we did not evaluate substances secreted from Malassezia, it is known that secreted proteases or metabolites from Malassezia could damage skin barrier [1,2]. We believe this may cause more barrier defects and less efficacious function of the new drugs in restoring barrier defects and inflammatory molecules in Malassezia+cytokines-treated groups compared to Malassezia-only treated groups. We added this explanation in the discussion section.

[1] Prohic, A., Jovovic Sadikovic, T., Krupalija‐Fazlic, M., & Kuskunovic‐Vlahovljak, S. (2016). Malassezia species in healthy skin and in dermatological conditions. International journal of dermatology, 55(5), 494-504.

[2] Harada, K., Saito, M., Sugita, T., & Tsuboi, R. (2015). Malassezia species and their associated skin diseases. The Journal of dermatology, 42(3), 250-257.

Point 4: 2.2) In figure 2 is possible to observe that IL4 and 13 affects the production of IVL and Malassezia did not. In this sense, the utilization of this drug is once more not pointing for an interesting result regarding Malassezia.  The same I observed for LOR. Can the authors clarify why this drug should be important in the context of Malassezia?

Response 4: We greatly appreciate the reviewer’s insightful comments. As you pointed out, IVL expression is not affected by MR itself, but MR inhibits the restorative effects of anti-IL-4Rα and ruxolitinib on IVL expression in the AD microenvironment. In the IL-4/IL-13 treated RHE, the skin barrier was already greatly collapsed, so we could not find additional barrier disruption by MR. Meanwhile, the expression level of the loricrin in the MR group decreased compared to the CTL. Moreover, the expression level of loricrin in the MR(+) anti-IL4Rα treated group was lower than the MR(-) anti-IL4Rα-treated group. It can be interpreted that MR affects the restoration efficacy of the drugs on skin barrier molecules in the AD environment by activating JAK/STAT pathway. Therefore, it is meaningful to examine the effect of drug treatment in AD in the presence of MR. We added this content in the main text.

Point 5: In the conclusions the authors have shown that Malassezia can reduce the skin barrier function and the treatment with this medicament is not effective or that the Malassezia infection can worse during the treatment? Please clarify this.

Response 5: Thank you for the comments. What we are trying to say is that ruxolitinib and anti-IL4Rα treatments are less effective in the presence of Malassezia, though the drugs are still effective in treating exacerbated AD. The conclusion has been clearly revised in light of your feedback.

Point 6: According to the aims, the authors have declaired: In this study, we investigate how M. restricta affects the AD model in 3D RHE and the effectiveness of ruxolitinib and anti-IL4Rα with respect to the skin barrier function and inflammation. I am confused regarding the relation of inflamamtion, Malassezia and the purposed drug. Please better clarify this in the present work. What is the central message addressed from your paper?

Response 6: We appreciate the valuable comments.

We apologize for the confusion in interpreting the results regarding the relationship between inflammation, Malassezia, and the drugs. Most Th2- and Th17-inflammatory markers increased in Malassezia-treated groups compared to Malassezia-non-treated groups with the same conditions (Fig. S4). VEGF, which is related to vascular permeability, was also affected by Malassezia. Interestingly, anti-IL4Rα treatment significantly reverted the changes induced by Malassezia together with IL-4 and IL-13 ; however, changes were not significant in the ruxolitinib-treated group. Anti-IL4Rα exhibited anti-inflammatory effects on major inflammatory pathways in the pathogenesis of AD, even in Malassezia overgrowth status, and it effectively decreased VEGF expression (Fig. 4). Thus, we assume that dupilumab-related facial redness is not directly caused by the drug, but it could be related to Malassezia overgrowth, inducing Th2 inflammation and VEGF expression. In the Malassezia-rich RHE, the anti-inflammatory activity of ruxolitinib was not enough in this study(Fig.4). Thus, previous studies which reported good therapeutic outcomes of ruxolitinib in the dupilumab-related facial redness may not be related to Malassezia overgrowth. Further studies are needed to reveal the therapeutic mechanism of ruxolitinib in treating dupilumab-related facial redness. We revised the relevant contents in the abstract, results and discussion, and conclusion section.

As stated in the conclusion, MR overgrowth could affect the skin barrier by increasing AMPs, deteriorating the lipid barrier, and stimulating the secretion of Th2 cytokines even in healthy skin models. Furthermore, in the AD microenvironment, MR amplified the secretion of AMPs, Th2, and Th17 cytokines, which activated the JAK/STAT signaling pathway and further reduced the production of skin barrier proteins, causing lipid barrier abnormalities. Therefore, treatment with anti-IL4Rα or ruxolitinib could be less effective in downregulating the levels of Th2- and Th17-inflammatory markers and restoring skin and lipid barriers in the presence of MR.

Our findings suggest that MR can exacerbate AD in both skin barrier and inflammation as well as decreasing the effectiveness of treatment with ruxolitinib and ant-IL4Rα.

Point 7: Figure 7 should not be a figure but a table.

Response 7: Thank you for the comments. We changed Figure 7 to Table 1.

Point 8: The authors contributions are unclear.

Response 8: Thank you for the comments. We changed the authors' contributions more clearly.

Round 2

Reviewer 1 Report

The authors reply all my questions. This article can be accepted by IJMS.

Reviewer 2 Report

All the observations included by the authors are fine.